# Polymer Composite Fabrication Reinforced with Bamboo Fiber for Particle Board Product Raw Material Application

**DOI:** 10.3390/polym13244377

**Published:** 2021-12-14

**Authors:** Martijanti Martijanti, Sutarno Sutarno, Ariadne L. Juwono

**Affiliations:** 1Department of Physics, Faculty of Mathematics and Natural Sciences, Universitas Indonesia, Depok 16424, Indonesia; martijanti@lecture.unjani.ac.id; 2Department of Mechanical Engineering, Faculty of Engineering, Universitas Jenderal Achmad Yani (UNJANI), Bandung 40274, Indonesia; 3Department of Metallurgy Engineering, Faculty of Engineering, Universitas Jenderal Achmad Yani (UNJANI), Bandung 40274, Indonesia; sutarno@lecture.unjani.ac.id

**Keywords:** polymer, composite, bamboo fiber, particleboard, mechanical properties

## Abstract

Bamboo particles as reinforcement in composite materials are prospective to be applied to particleboard products in the industry. This study aimed to synthesize bamboo particle reinforced polymer composites as a substitute for particleboard products, which still use wood as a raw material. The parameters of the composite synthesis process were varied with powder sizes of 50, 100, and 250 mesh, each mesh with volume fractions of 10, 20, and 30%, matrix types of polyester and polypropylene, Tali Bamboo, and Haur Hejo Bamboo as reinforcements. Characterization included tensile strength, flexural strength, and morphology. Particleboard products were tested based on JIS A 5908-2003, including density testing, moisture content, thickness expansion after immersion in water, flexural strength in dry and wet conditions, bending Young’s modulus, and wood screw holding power. The results showed that the maximum flexural and tensile strength values of 91.03 MPa and 30.85 MPa, respectively, were found in polymer composites reinforced with Tali bamboo with the particle size of 250 mesh and volume fraction 30%. Particleboard made of polypropylene and polyester reinforced Tali Bamboo with a particle size of 250 mesh and a volume fraction of 30% composites meets the JIS A 5908-2003 standard.

## 1. Introduction

Natural fiber-reinforced polymer composites are an affordable and available natural alternative to wood with potential application in the automotive industry, construction field, structural components, furniture, and other products [1]. The selection of the composite manufacturing process depends on the type of fiber, the volume fraction of the matrix, or the fiber material because each material has different physical and mechanical properties. Composites have also shown remarkable resistance to impact, wear, corrosion, and chemicals, but these properties depend on the composition of the material, the type of fiber, and the type of manufacturing technique used to produce them. The research development will be more directed towards finding new composite structures with different combinations of variants and adopting new manufacturing techniques [2].

There are many types of natural fiber-producing sources from plants such as flax, straw, wood, rice husks, wheat, barley, wheat, rye, sugar cane (sugar and bamboo), grass, reeds, kenaf, hemp, oil palm, empty fruit bunches, sisal, coir, water hyacinth, kapok, mulberry paper, raphia, banana fiber, pineapple leaf fiber, and papyrus [3]. Among them, bamboo fiber is one of the most promising substitutes to synthetic fiber. In addition to its strength and stiffness with low density, it is also inexpensive, has a short growth cycle, it is readily available, environmentally friendly, highly flexible, easy to develop, and biodegradable [4].

Bamboo as a source of fiber is estimated to have more than 1250 species, with world production reaching 10 × 10^6^ tons. It is the third-largest production after wood and cotton, with bamboo fiber prices reaching USD 0.5–1/kg. Indonesia is the second-largest exporting country, with a value of USD 269 million, following China’s USD 1 billion thirty-four million. Bamboo production in Asian countries reached 65% of global production, followed by the U.S. 28% and Africa 7%. The major bamboo importers were Japan, worth USD 194 million, the U.S., worth USD 254 million, and Europe, worth USD 230 million [5].

The chemical components of bamboo are mainly cellulose (55%), hemicellulose (20%), and lignin (25%) [6]. Mechanically, bamboo fiber has high tensile strength (140–800 MPa) and a high modulus of elasticity (33 GPa) with a low density of 0.6–0.8 g/cm^3^ [7]. The specific properties of bamboo fiber, such as low specific gravity, specific strength, and stiffness, are comparable to glass fiber [8,9,10,11].

Natural particle-reinforced composites, especially bamboo particles, obtained composite tensile strength values of 1.53–7.50 MPa and Modulus of Elongation (MOE) 0.38–1.26 GPa. The optimum values in this study are the particle size of bamboo in 40 mesh with a particle volume fraction of 50%, noting that the size and volume fraction of the particles will affect the tensile strength, MOE, and Modulus of Rupture (MOR) of the bamboo-PVC composite particle [12].

Unidirectionally reinforced particle composite was made of bamboo particle bundle and Bamboo powder. The results showed that the tensile strength and flexural strength of the bamboo powder increased with increasing unidirectional particle content. The highest tensile and flexural strengths were 169.9 MPa and 221.1 MPa, respectively [13]. Tensile and flexural strength values of the bamboo powder composite ranged from 1.7–14.3 MPa and 31.3–79.8 MPa, respectively [14]. The tensile and flexural strengths of unidirectional bamboo particles ranged from 23.1–169.9 MPa and 130.1–221.1 MPa, respectively [15]. Bamboo length ranges from 23.5 to 33 MPa and 106.5 to 140.1 MPa [9,13,14,15,16].

Bamboo particle composites were added to the polyester matrix [17]. By varying the volume fraction for bamboo particles (15, 30, 45)% and alumina particles (0, 5, 10, 15)% using a polyester matrix, the resulting tensile strength of the composite increased along with the number of particles and alumina fractions. In a study by Djamil, a composite consisting of an epoxy polymer matrix with Gigantoacloa Apus bamboo particles, in the form of a laminate, using the hand layup method, yielded a strength of 11.05 ± 0.55 MPa and tensile strength of 15.43 ± 1.14 MPa [18].

Several studies have been conducted on the effect of particle volume fraction on the mechanical properties of bamboo particle composite materials, both pure bamboo particles and a combination of 2 different types of particles [13,19,20,21,22], types of thermosetting and thermoplastic matrices [18,22,23,24], fiber shape and size [25,26,27], and fiber orientation [28,29,30], all of which target an increase in mechanical properties.

Bamboo particles were used as an additive in CBC (Cork Bamboo Composite), resulting in increased tensile strength, dimension stability, and air absorption. In this study, the optimum value is reached by adding 10% bamboo particles. The resulting CBC product can be applied as an alternative material for flooring and structural application [31]. Research on the use of bamboo particles to meet the mechanical and physical characteristics of the particleboard aligned with the industry standards is ongoing. A number of studies are currently evaluating the effects of varying percentages and sizes of bamboo particles, as well as the factor of additives between the fiber and the matrix, and the type of matrix and fiber [32,33,34,35].

Previous studies have focused on the factors in the manufacturing of composites, such as the particle volume fraction, matrix type, and particle size, which directly affect the tensile and flexural strength of the composites. In this study, bamboo species, volume fraction, and particle size were analyzed with respect to the flexural strength and tensile strength of polymer composites reinforced with bamboo particles applied to particleboard. The particleboards were tested according to the JIS A 5908-2003 standard.

## 2. Materials and Methods

### 2.1. Materials

#### 2.1.1. Bamboo Particles

The selection of bamboo was obtained from a bamboo plantation located at 500 m above sea level in City Bandung, West Java, Indonesia, and the gathering crops were carried out in the middle of the dry season (June–September). The bamboo was cut into slats of 1 mm thickness (0.2–0.5) mm and a length of 300–500 mm. It was then processed in a crushing machine to produce small pieces. The bamboo types Tali Bamboo (*Gigantochloa apus*) and Haur Hejo Bamboo (*Gigantochloa atter*) were used, and they were identified as SBT for Tali Bamboo and SBH for Haur Hejo Bamboo. For SBT, the alkalization was conducted in a 6% NaOH solution in an autoclave with a pressure of 0.5 MPa at a temperature of 120 °C for 2 h. For SBH, the alkalization was conducted in 4% NaOH solution in an autoclave with a pressure of 0.5 MPa at a temperature of 120 °C for 3 h. The bamboo particles were rinsed in water at a temperature of (90–95) °C and dried in an oven at a temperature of 105 °C for 10 h, followed by washing with acetic acid. The SBT and SBH bamboo particles were prepared into 50, 100, and 250 mesh particle sizes.

#### 2.1.2. Polymer Matrix


Polypropylene homopolymer from Binhson Refining and Petrochemical Co., Ltd. (Hanoi, Vietnam) was used. A mixture of coupling agents was added. The composition and ratio were as follows: 2% benzoyl peroxide (C_14_H_10_O_4_) + 9% maleic anhydride (C_4_H_2_O_3_) + 89% polypropylene.Polyester consists of two components, resin, and hardener. The polyester type was Yukalac 157 BQTN-EX orthophthalic matrix and Mekpo (Methyl ethyl ketone peroxide) hardener from PT Bratachem Bandung (Bandung, Indonesia).


### 2.2. Processing of Composite

Fabrication of polymer composites reinforced with bamboo particles started with the manufacture of composite samples for tensile and flexural testing carried out by hand layup and extrusion methods; then particleboard fabrication was carried out. Processing of composite is as shown in Figure 1 and Figure 2. Composite fabrication and characterization process parameters are shown in Table 1.
1.Bamboo Particle Reinforced Polyester (BPRP) composite fabrication

Preparation of polyester matrix by mixed polyester resin and hardener in a ratio of 100:1 Mixture was stirred at low speed for 1 min at room temperature. This resulted in the mixed polyester matrix with volume fraction variation of 10, 20, and 30% for each SBT or SBH particle size. Pouring a mixture of SBT or SBH fiber with a polyester matrix into the sample mold was performed with a pressure of 0.05 MPa for 30 min at room temperature; then, a vacuum process was carried out, and the mixture was cured for 24 h.
2.Bamboo Particle Reinforced Polypropylene (BPRPP) composite fabrication

The BPRPP composite fabrication involved two stages:

Stage 1: Bamboo particle, polypropylene, coupling agent, and lubricant were mixed until homogeneous in a mixer, and the temperature reached 115 °C, then cooled to 50 °C. The product was in a pellet.

Stage 2: The pellet was processed through a single screw extrusion machine with a barrel temperature of 210–220 °C for molding. The thickness of the composite sheets was calibrated on a roll calibrator, and the composite sheets were cooled on a roller conveyor.

### 2.3. Composite Testing

#### 2.3.1. Tensile and Flexural Strength

The bending strength of the composite Tali bamboo particle 50, 100, and 250 mesh using a matrix of polyester and polypropylene was based on ASTM D 638 and ASTM D 790 standards. The tensile test sample shown in Figure 3 is mounted on a Hung Ta Instrument Go.Ltd, type HT-8503 tensile testing machine, with a crosshead speed of 10 mm/min, capacity 100 kN.

#### 2.3.2. SEM Testing

To identify the effect of the fiber extraction process on morphology, SEM testing was carried out using the JEOL JSM-6360, 15 kV with magnifying of 1500 times. Compilation of process parameters and test results data such as tensile strength, identification of functional groups, and morphology was then analyzed and evaluated.

#### 2.3.3. Bamboo Fiber Reinforced Polymer Composite Particle Board Testing

Particleboard testing was carried out based on the JIS A 5908-2003 standard, which included testing for density (g/cm^3^), moisture content (%), swelling in thickness after immersion in water (%), flexural strength in dry and wet conditions (MPa), tensile strength, and wood screw holding power.

## 3. Result and Discussion

### 3.1. Mechanical Properties

The effect of the particle size of bamboo 50, 100, and 250 mesh, and the volume fraction of its particles 10, 20, and 30%, on the flexural strength of PP and UPR, are shown in Figure 4 and Figure 5, respectively. The increase in the size and volume fraction of the bamboo particles is proportional to the increase in the flexural strength of polymer composites (polyester, polypropylene) reinforced with Tali Bamboo and Haur Hejo Bamboo fiber. This is because the strength of the composite will be influenced by the number of fibers used in the composite fabrication process; the increasing number of fibers will be directly proportional to the increase in the strength of the composite. This is in accordance with the rules of the Rule of Mixtures, which states that the strength of the composite is influenced by the strength and volume fraction of the fiber and matrix.

Research on the characteristics of the bamboo fiber-reinforced composite material was focused on the granular size of the bamboo particles (20, 40, 60) mesh, the volume fraction of bamboo fiber, and Polyvinyl Chloride (PVC) matrix with variations in the fiber fraction (30, 40, 50)%, and the process of making composites with the hot press with processing parameters for the pressure of 6 MPa, pressing temperature of 448.15 K with pressing time of 8 min. The combination bamboo particle size of 40 mesh with fiber volume fraction of 50% gives the optimum strength value [12]. In fiber in the form of particles, the smaller the particle size of the fiber, the wider fiber dispersion on the surface of the matrix, resulting in the matrix binding the fiber particles well. The adhesion force bond is also stronger than the cohesion force, resulting in an increase in the strength of the composite [36].

Another factor that affects the strength of the composite is the interfacial bond between the fiber and the matrix, which plays an important role in determining the mechanical properties of the composite. Since stress is transferred between the matrix and fiber across the interface, solid interfacial bonding is required to achieve optimal reinforcement, although it is possible to have a strong interface while allowing crack propagation (which can reduce toughness and strength). However, for natural fiber-reinforced composites, there is usually limited interaction between the hydrophilic fibers and the generally hydrophobic matrix leading to poor interfaces. For a good bond between the fiber and the matrix, treatment of the fiber must be carried out to increase the interfacial strength of the matrix and fiber. This will affect the toughness as well as the tensile and flexural strength of the composite material [37]. Thus, the bamboo powder used in this study was subjected to an alkalization process to increase the bond quality between the fiber and the matrix.

The effect of bamboo species on the flexural strength of composites on SBT-reinforced polymer composites, on average, gives a greater value than SBH-reinforced polymer composites because the strength of the composite is influenced by the strength of the fiber alone; the degree of the fiber strength directly impacts the degree of the strength of the composite). The results of the first stage of the study on fiber alkalinization showed higher SBT fiber strength than SBH fiber strength, and this influenced the strength value of the composite. Fibers are usually stronger and stiffer than the matrix; the strength and stiffness of composites are generally seen to increase with increasing fiber content [37,38,39,40].

The flexural strength of the polyester–SBT composite is in the range of values 29.52–46.89 MPa, the polypropylene–SBT composite is in the range of values 38.81–91.03 MPa, and the flexural strength of the polyester–SBH composite is in the range of values 27.97–49.71 MPa, polypropylene-SBH composites are in the range of values 33.41–60.82 MPa. The effect of the matrices (polyester, polypropylene) on the flexural strength of the SBT and SBH-reinforced composites was evaluated, showing that the polypropylene matrix has a greater value in use compared with polyester.

It is a natural property of the matrix that it can bind to the fiber, and the polypropylene matrix has a very good bond with bamboo particles. Between the polypropylene and bamboo particles in the composite-making process of mixing solid raw materials in the form of pellets, the matrix is able to wet the substrate easily with the adhesion force greater than the cohesion, resulting in easier absorption of molecules. In addition, the strength of the polypropylene matrix is greater than that of the polyester matrix because the strength of the composite depends on the strength of the matrix that forms the composite material, where the strength of the polypropylene matrix is 30.62 MPa, and the polyester matrix is 19.44 MPa.

The flexural strength of polymer composites (polyester, polypropylene) with SBT and SBH has increased, compared with the pure matrix, with the flexural strength of polyester–SBT composites ranging from 52–144% and polyester–SBH ranging from 44–156%. The flexural strength of the polypropylene–SBT composites ranged from 27–197%, and the polypropylene–SBH ranged from 9–99%. 

The increase in strength in the composite is dependent on the fibers; the load imposed on the composite is initially received by the matrix and then transmitted to the fiber so that the fiber will withstand the load until the maximum load is reached. The fiber used is bamboo fiber in particle size, resulting in a more uniform strength in all directions. The smaller the particle size, the easier it will be for them to be evenly distributed throughout the matrix, resulting in a stronger bond between the particles and the matrix. This results in the transfer of tension between the matrix and bamboo particles which increases the flexural strength and consequently the strength of the entire composite.

The flexural strength of particleboard based on JIS A 5908-2003 standard is (8–30 MPa), polymer composites (polyester, polypropylene) with SBT and SBH meet the standards set by JIS, namely the flexural strength of polyester–SBT composites is in the range of values (29.52–46.89) MPa. The polypropylene–SBT composite was in the range of values (38.81–91.03) MPa, and the flexural strength of the polyester–SBH composite was in the range of values (27.97–49.71) MPa. The composite polypropylene–SBH is in the range of values (33.41–60.82) MPa.

The results of the bending test on composites by varying the type of matrix, type of bamboo, size, and volume fraction of fiber obtained show maximum conditions in polyester–SBT and polypropylene–SBT composites, with a volume fraction of 30% and bamboo particle size 20 mesh, and can be seen in Figure 6 and Figure 7 respectively.

The graph of the stress–strain relationship for bending testing on polyester–SBT has a breaking stress value of 3.561 MPa, which is smaller than the breaking stress of polypropylene–SBT of 5.594 MPa. In the graph of the test results for the polyester–SBT composite, it can be seen that the material does not undergo a necking process but immediately breaks in the maximum area, while on the graph of the test results for the polypropylene–SBT composite when the stress reaches the maximum value, then necking occurs and finally breaks. The results of the bending test on the polyester–SBT composite were calculated mathematically for a bending strength of 47.67 MPa with a bending elasticity modulus of 3.04 GPa. In comparison, the bending strength of a polypropylene–SBT composite was 90.70 MPa with a bending elasticity modulus of 3.93 GPa. Polypropylene–SBT composites have stronger and more elastic characteristics than polyester–SBT composite, with the same volume fraction factor and particle size. Different types of matrices will affect the strength of the composite because it is influenced by the strength of the matrix. The greater the strength of the matrix, the greater the strength of the composite [9,13,14,15,16].

Bamboo fiber-reinforced composites have a higher flexural strength value than bamboo fiber particle-reinforced composites, as shown in Figure 8. In the current study, the SBT and SBH polyester–particle composites have a minimum flexural strength value of 29.52 MPa, respectively, and 27.97 MPa, while the maximum values were 46.89 MPa and 49.71 MPa, respectively. The polypropylene–particle composite SBT and SBH had minimum flexural strength values of 38.81 MPa and 33.41 MPa, respectively, while the maximum values were 91.03 MPa and 60.82 MPa, respectively. Other studies have shown a minimum value of 26.23 MPa and a maximum value of 79.86 MPa. This indicates that the minimum and maximum flexural strength values in this study have higher and lower values compared with previous studies but are still within the range of values from previous studies.

The effect of bamboo particle size 50, 100, and 250 mesh and volume fraction of bamboo particles 10, 20, and 30% on tensile strength of PP and UPR are shown in Figure 9 and Figure 10, respectively. The increase in size and volume fraction of bamboo particles is proportional to the increase in tensile strength of polymer composites polyester (UPR), polypropylene (PP) reinforced with both SBT and SBH.

The effect of the type of bamboo on the tensile strength of the composite did not give a significant increasing effect; the tensile strength of the polyester–SBT composite was in the range of values 12.62–24.06 MPa, the tensile strength of the polyester–SBH composite was in the range of values 16.34–27.40 MPa. The average tensile strength of the SBT-reinforced polymer composite has a higher value than the average tensile strength of the SBH-reinforced polymer composite.

In a previous study, the process of making composites consisted of varying the composition of the fiber and matrix, with a composition of (30, 50, 70)% fiber by obtaining the highest value at 70% bamboo fiber composition for tensile strength of 265 MPa and modulus elasticity of 12.4 GPa; theoretically the larger the volume fraction used, higher the strength value [14]. In other studies, the number of fibers used in the composite manufacturing process shows an effect on increasing the tensile strength value by 0, 4, 8, 12, and 16%; the greater the volume fraction of the fiber used, the greater the tensile strength of the composite [41]. Research is also ongoing on the shape of the fibers used (fiber, powder) in various percentages (0, 50, and 100) and the function of bamboo both as fiber and matrix.

In a previous study, the temperature parameters were varied (160 °C, 180 °C, and 200 °C) at a pressure of 65 MPa, resulting in the value of the flexural modulus and tensile strength increasing linearly with increasing fiber content [14]. The design of the composite manufacturing process was carried out by varying the fiber volume fraction and combining natural fibers (bamboo) and alumina fibers. The volume fraction for bamboo fiber (15, 30, and 45)% and alumina fiber (0, 5, 10, and 15)% using a polyester matrix gave the results that the tensile strength of the composite increased along with the number of fiber and alumina fractions added to the polyester matrix [17]. The increase in strength of the composite is dependent on the strength of fibers used; the load imposed on the composite is initially received by the matrix and then transmitted to the fiber. Thus it is the fiber that withstands the load until the maximum load is reached. The fiber used was bamboo fiber in particle size, resulting in a more uniform strength in all directions. The smaller the particle size, the easier it is for particles to be evenly distributed throughout the matrix. The stronger the bond between the particles and the matrix that causes the transfer of tension between the matrix and bamboo particles, the higher the tensile strength and the strength of the composite.

The tensile strength of polymer composites (polyester, polypropylene) with SBT and SBH increased compared with pure polymer without bamboo particles. The SBT–polyester composites ranged from 28–144%, polypropylene–SBH composites ranged from 7.65–55%. The SBH–polyester composites ranged from 65–165%, and polypropylene–SBH composites ranged from 1.71–42%. This increase occurs because the bamboo particles act as reinforcement which distributes the tensile load to the matrix so that the tensile strength increases. The tensile strength of the composite is consistent with previous researchers such as [18,19,22,23,42]. When the matrix is reinforced with bamboo particles, this tensile strength is significantly increased, where the SBT or SBH particles resist the load and consequently increase the strength of the composite material. The tensile strength increases with the addition of SBT or SBH percentage particles, where the SBT or SBH particles are dispersed over a large area in the matrix [37].

The size and volume fraction of SBT and SBH particles have an effect of increasing the tensile strength of the composite. The results of this composite tensile strength study are aligned with the research conducted by Huang, namely 1.53 MPa at a particle size of 20 mesh and 7.50 MPa at a particle size of 40 mesh. The results of Huang’s research indicate that the optimal size of bamboo is 40 mesh because the size of the bamboo will affect the tensile strength of the composite reinforced by bamboo fiber. The finer particle size of the bamboo will help increase the tensile strength of the composite because the smaller particle size causes the more comprehensive bonding with the particles on the surface of the matrix, making it possible for the accepted load to be even greater [37].

The results of the tensile strength test obtained the maximum value for the polypropylene–SBT composite with a 30 vf% and the bamboo particle size of 250 mesh, then compared with the polyester–SBT composite at the condition that the volume fraction and particle size of bamboo were fixed. The graph of the relationship between the stress and strain of the polypropylene–SBT and polyester–SBT composites can be seen in Figure 11 and Figure 12, respectively.

The tensile strength of the polypropylene–SBT composite of 32,430 MPa has a higher value than the tensile strength of the polyester–SBT composite of 24,259 MPa because of the strength of the composite is influenced by the value of the tensile strength of the matrix. The tensile strength of pure polypropylene has a greater value than the tensile strength of pure polyester, so it has an effect on increasing the value of the tensile strength of the composite [9,13,14,15,16].

The results of the tensile strength test of polypropylene–SBT show that the behavior of the material has strong characteristics while maintaining its elastic properties. On the graph, the stress increases linearly with the addition of the elongation value of the material until it reaches maximum strength, then a new necking process occurs. Polyester–SBT composites show a different behavior; in the graph of stress to strain, there is no necking, but after reaching the maximum stress, it immediately breaks.

The comparison between the graph of the tensile and the bending test results for the condition of the material with the type of bamboo, particle size, and volume fraction still give the same material behavior (see Figure 6, Figure 7, Figure 11 and Figure 12). The relationship between stress and strain from tensile test and bending test results on SBT–polyester composites gives similar profiles; namely, the stress increases linearly with increasing length, and there is no necking behavior but immediately breaks. This occurs in the graph of the relationship between stress and strain from the tensile and bending test results on the polypropylene–SBT composite; namely, the stress increases linearly with the increase in length until reaching the maximum when necking and fracture occur. This relationship has been examined in previous research [43].

Tensile test results from previous researchers for polymer composites with bamboo fiber are presented in Figure 13. Based on Figure 13, polymer composites (polyester, polypropylene) reinforced with SBT and SBH show minimum tensile strength values for polyester–SBT composites 12.62 MPa, polyester–SBH composites 16.34 MPa, polypropylene–SBT composite, and 20.20 MPa polypropylene–SBH composite. The respective maximum values were for polyester–SBT composite 24.06 MPa, polyester–SBH 40 MPa, polypropylene–SBT 30.85, and polypropylene–SBH 28.22 MPa. Other studies have reported a minimum value of 1.70 MPa and a maximum value of 15.60 MPa.

Compared with composites in the form of fibers, the tensile strength of composites in the form of particles in this study is still lower, namely the tensile strength of composites in the form of fibers has a minimum value of 74 MPa and a maximum value of 169.9 MPa. The tensile strength of SBT and SBH-reinforced polymer composites in the form of particles has a higher strength value than the tensile strength of bamboo particles in previous studies but lower than the tensile strength of bamboo fiber because the strength of the composite is influenced by the length of the fiber, the increase in size fiber length is proportional to the tensile strength of the composite.

### 3.2. Bamboo Fiber Reinforced Polymer Composite Morphology

The correlation between particle size variations of 50, 100, and 250 mesh bamboo particles, at fiber volume fractions of 10, 20, and 30% and the dispersion and matrix wetting characteristics of bamboo powder was observed by SEM, revealing that bamboo powder (SBT and SBH) were well wetted by the matrix polymers (polyester, polypropylene) and the same is seen for all composite samples, as shown in Figure 14, Figure 15, Figure 16 and Figure 17.

Figure 14 and Figure 15 explain that the sample between bamboo powder and polyester matrix shows the interface of the fine slices showing the dispersion of bamboo particles in the matrix and the possible compatibility between bamboo and matrix. The particle–matrix compatibility affects the high interfacial strength because the bamboo particles are well wetted by the matrix. In addition, there is a proton donor-acceptor interaction between the matrix and the bamboo lignin chain and the interaction between the hydroxyl or carbonyl groups of bamboo lignin and the -H matrix [12]. Figure 14 and Figure 15 indicate that the cross-sectional flexural interface is sharp and indicates that some of the bamboo particles have been withdrawn from the resin matrix and reveals a possible weak bond between the bamboo particles.

The study of natural fiber composites observed deflections or changes in crack paths with changes in orientation and fiber volume fraction. Riverline fractures are quite visible on the surface of the matrix, as shown in Figure 14. It was observed that the presence of bamboo particles in the polyester matrix, which has a brittle nature, slightly changes the overall properties from completely brittle to slightly ductile. The presence of fibers in the matrix helps form a barrier that disrupts the streamlines on the surface of the brittle matrix [44].

The morphology, as shown in Figure 14, is supported by tensile strength and bending strength data when compared to the condition of the pristine matrix. The strength value of the matrix is lower than the composite. The comparison of results from the tensile strength and bending strength test showed that the composite’s tensile strength and bending strength increased by 28–165% and 27–197%, respectively (Figure 4, Figure 5, Figure 9 and Figure 10). The stress and strain graph shows a straight line, as shown in Figure 11, which shows the elasticity of the behavior of the polyester–SBT composite. A change from the behavior of a brittle material (marked by river line fractures) to a ductile material. The debonding occurs between the fiber and the matrix because the fiber is embedded deep in the matrix, indicated by the yellow arrow in Figure 10 (UPR–SBT for 250 mesh, 30 vf%). This was done in a previous study that examined the morphological behavior and mechanical properties of bamboo fiber reinforced polymer composites [44,45,46,47,48].

The river line fractures seen on the matrix surface in Figure 14 (UPR–SBT, 50 mesh, 20 vf%) and 14 (UPR–SBT, 50 mesh, 30 vf%) are characteristic of brittle fractures in the polyester matrix, which are marked by yellow arrows. These lines have been widely observed by previous researchers [44,49,50]. The fiber-reinforced composites will be randomly distributed, resulting in the phenomenon of breakage of the fibers and the occurrence of debonding between the fibers and the matrix, because the fibers are embedded deep in the matrix, this can be seen in Figure 14 and Figure 15, marked by a line yellow arrow. When stress is transferred from the polyester matrix to the fibers, fiber breakdown occurs instead of debonding or pulling out. The same thing has been explained by previous researchers [51] the effect of filler particles on the fracture of the composite.

Figure 14 and Figure 15 depict several hollow structures in composite fabrication in the presence of an imperfect vacuum process, providing an opportunity to form microbubbles and microholes to form on solidification, as reported by the previous research process [38]. The bamboo fiber particles are fully embedded and covered by polyester, having good homogeneity and minimum porosity. The fiber tension shows that there is a perfect match between the bamboo and polyester fibers. The fibers in this study underwent an alkalizing immersion process and added a pre-impregnated matrix, showing that the fibers and matrix traces still surround the matrix to the fiber. This shows that it has not been fully able to penetrate the fiber so that the fiber and matrix interface increases [5,38].

Figure 16 and Figure 17 reveals that the bamboo particles are well moistened by the polypropylene matrix; the same is seen for all composite samples. This happens because the presence of the MA-g-PP carboxyl group will have an effect on better quality for the wetting of the matrix (polypropylene) to the bamboo particles. Figure 16 and Figure 17 illustrate the cleavage of the fiber bundle or tears in the cracked plane. This observation indicates that there is a fairly good interfacial adhesion of fibers between the matrix and the bamboo particles, reflecting the presence of a small observed void content, as reported in a previous study [2]. The polypropylene matrix modified by maleic anhydride can be utilized to improve the interaction of the hydroxyl groups on bamboo fiber, which will improve the mechanical properties of the composite [52].

Cracks are usually seen when there is a strong bond between the fiber and the matrix; when the fiber is pulled due to stress transfer, then the matrix around the fiber becomes cracked. The same phenomenon has been described by Khan et al. [44] in their work on the fiber-matrix debonding at the interface and by Johnson et al. [53] in their work on the roles of the matrix cracking and fiber-matrix debonding on stress transfer between the fiber and the matrix in single fiber fragmentation assays. Johnson et al. [53] adhere to this matrix by a strong interface between the fibers and the matrix resulting in transverse cracking of the matrix. Figure 17 shows a path that has been blocked or deflected by the presence of fiber. Crack pinning or crack path deflection is the same phenomenon as described by [54,55] during their work on flax fiber composites, where they observed the deflection or crack path change with changes in fiber orientation and fiber volume fraction.

The interfacial morphology of PP-SBT and PP–SBH by studying the tensile crack surface under SEM is shown in Figure 16 and Figure 17. It turned out that the bamboo particles were embedded in the PP matrix, and a tubular structure was formed, maleic anhydride exerting a filling effect between them. Figure 16 and Figure 17 show the infiltration of PP into the pores of the bamboo particles; this mechanical interlocking also increases the tensile properties of the better composite. This is supported by an increase in the tensile strength value of PP–SBT and PP–SBH, compared to pristine PP, which experienced an increase in tensile strength of 27–197%, as shown in Figure 9 and Figure 10. The stress and strain graph shows the behavior of the line linear results in better elasticity of the material, supported by the morphology in Figure 16 (PP-SBT, 250 mesh, 30 vf%), in accordance with research [45,48,56,57,58].

Figure 17, for FBH bamboo particles with polypropylene seen cracking into the strong interface between fiber and matrix resulting in transverse cracks in the matrix, the crack path was blocked or deflected by the presence of the bamboo particles. These crack path lines are similar to the phenomenon described by Keck [55].

### 3.3. Results Characteristics of Particle Boards Made from Bamboo Fiber Reinforced Polymer Composites

The fabrication of particleboard made from SBT–polypropylene composite is carried out through the extrusion process and is made from SBT–polyester composite using the hand layup method. The board testing was carried out based on the JIS A 5908-2003 standard with a total of 5 samples for each test, including density testing (g/cm^3^), moisture content (%), thickness expansion after water immersion (%), flexural strength under conditions dry (MPa), flexural strength after water immersion (MPa), bending Young’s modulus (GPa), and wood screw holding power (N). The test results are the average values and can be seen in Table 2.

The results of the density test on particleboard made from polypropylene–SBT composite are 0.879 g/cm^3,^ and polyester–SBT composite is 0.949 g/cm^3^, on polypropylene–SBT composites meet the JIS standard values of 0.4–0.9 g/cm^3^ while polyester–SBT exceeds JIS standard value. This density is influenced by the volume fraction percentage of the matrix and fiber; the polyester–SBT composite can meet the JIS standard if the fiber percentage is increased and the polyester percentage is reduced. JIS standard for moisture content (Wab) should not be more than 13% and swelling in thickness after immersion in water (Sw) is a maximum of 12%. Moisture content for polypropylene–SBT composites is 2.03%, and polyester–SBT is 1.21%. Swelling in thickness after immersion in water for polypropylene–SBT composites was 0.87%, and for polyester–SBT composite was 0.71%.

The bending strength of the dry condition of the particleboard composites, respectively, was 91.09 MPa for polypropylene–SBT and 46.89 for polyester–SBT. The bending strength in wet conditions of polypropylene–SBT composite is 46.82 MPa, and polyester–SBT is 24.27. The bending strength in dry and wet conditions is in the range of JIS standard values. The flexural modulus of elasticity of particleboard products made from polypropylene–SBT composite is 4.508 GPa, and polyester–SBT is 3.396 GPa. Test of wood screw holding power from polypropylene–SBT composite is 826 N and polyester–SBT composite is 740 N. This finding aligns with JIS A 5908-2003.

The test results for the characteristics of density, moisture content, thickness expansion after immersion in water, flexural strength in dry conditions, flexible strength after immersion in water, bending Young’s modulus, and wood screw holding power, have values following JIS Standard 5908-2003. Thus, polypropylene–SBT and polyester–SBT composite can be applied for particleboard products, as these composites have the characteristics of strong and elastic material. Testing the characteristics of the polypropylene–SBT composite particleboard product shows it has a higher test value than the polyester–SBT composite particleboard because the composite characteristics are influenced by the particleboard manufacturing method. In accordance with previous research, it was shown that the fabrication method (injection molding, extrusion, compression molding, hand layup) along with the composite manufacture process parameters and particle dispersion would affect the mechanical characteristics of the composite material [36,43,55,59].

## 4. Conclusions

Characterization of composite polymers (polyester, polypropylene) with reinforcement Tali and Haur Hejo bamboo particles form increased in tensile strength and flexural strength along with the increase in bamboo particles volume fraction and bamboo particle size was investigated. The results revealed that the maximum flexural and tensile strength values of 91.03 MPa and 30.85 MPa, respectively, belong to polymer composites reinforced with Tali bamboo particle size of 250 mesh and volume fraction of 30%. Morphological observations with SEM supported the improvement of the mechanical properties of bamboo composites. The particles were entirely embedded and covered by the matrix, had good homogeneity and minimum porosity.

Characterization and fabrication of board products made from polypropylene-Tali Bamboo and polyester-Tali Bamboo composites met the JIS A 5908-2003 standard. This provides opportunities and potential for polypropylene-Tali bamboo and polyester-Tali bamboo composites to be applied as one of the materials for particleboard products.

## Figures and Tables

**Figure 1 polymers-13-04377-f001:**
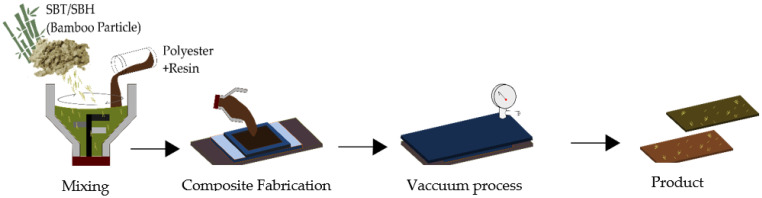
BPRP Composite Fabrication Process.

**Figure 2 polymers-13-04377-f002:**
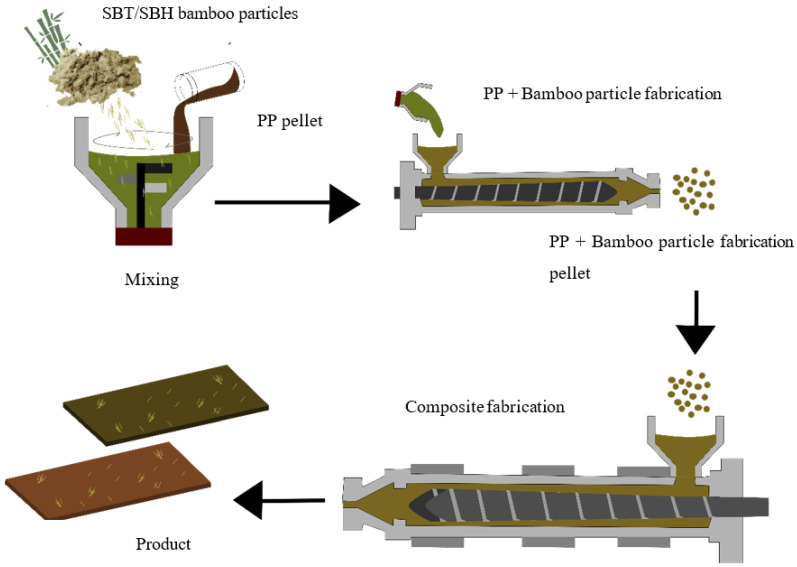
BPRPP Composite Fabrication Process.

**Figure 3 polymers-13-04377-f003:**
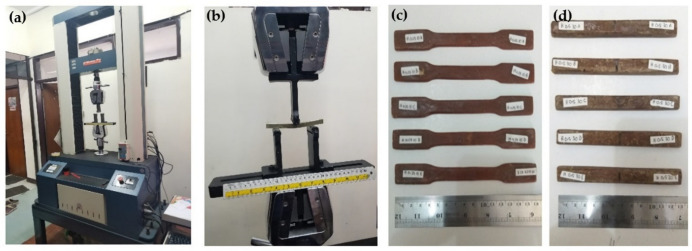
Preparation: (**a**) tensile test machine; (**b**) flexural test machine; (**c**) tensile test specimen; (**d**) flexural test specimen.

**Figure 4 polymers-13-04377-f004:**
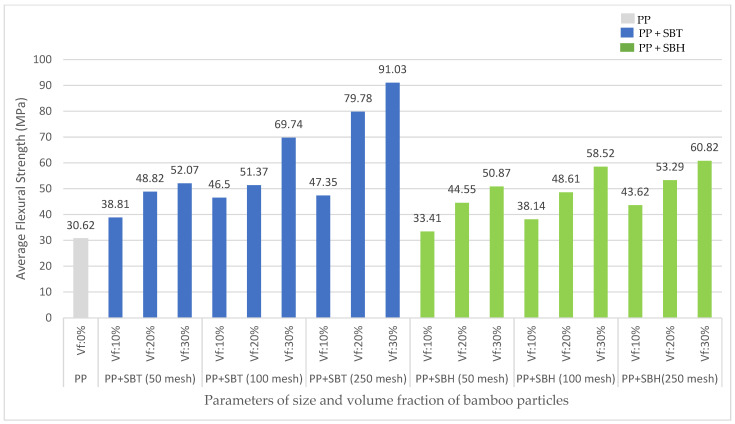
Relationship between PP composite flexural strength to volume fraction and particle size of SBT and SBH.

**Figure 5 polymers-13-04377-f005:**
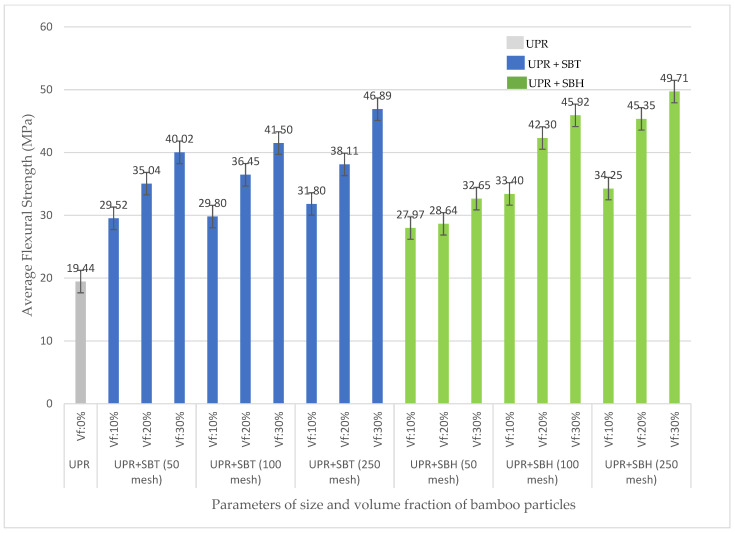
Relationship between UPR composite flexural strength to volume fraction and particle size of SBT and SBH.

**Figure 6 polymers-13-04377-f006:**
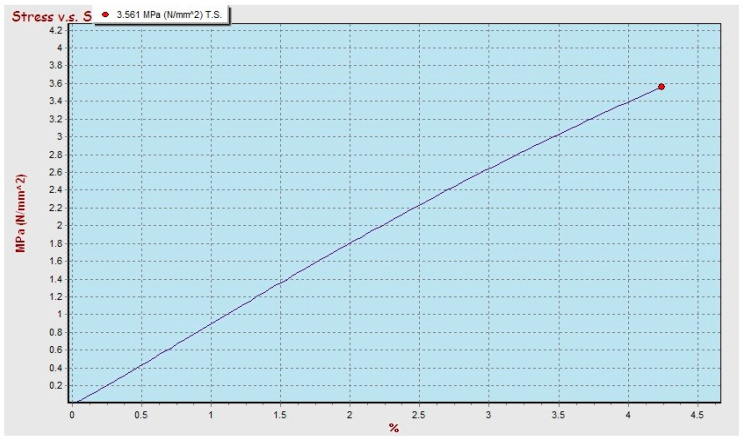
Stress-strain relationship on bending test of UPR–SBT composite with 30 vf% and 250 mesh particle size.

**Figure 7 polymers-13-04377-f007:**
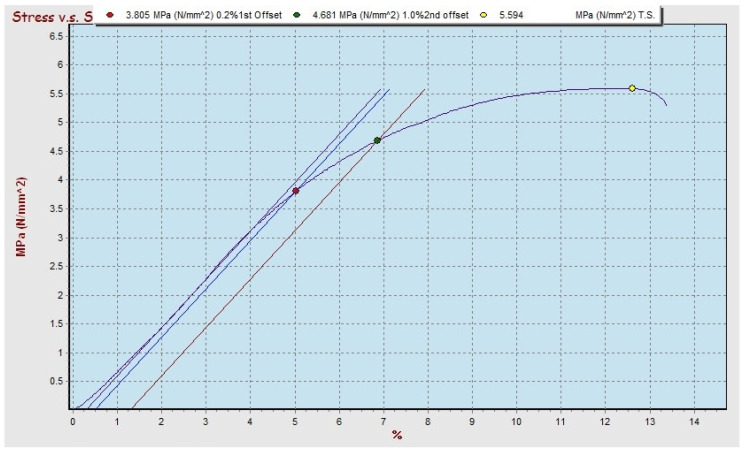
Stress-strain relationship on bending test of polypropylene–SBT composite with 30 vf% and 250 mesh particle size.

**Figure 8 polymers-13-04377-f008:**
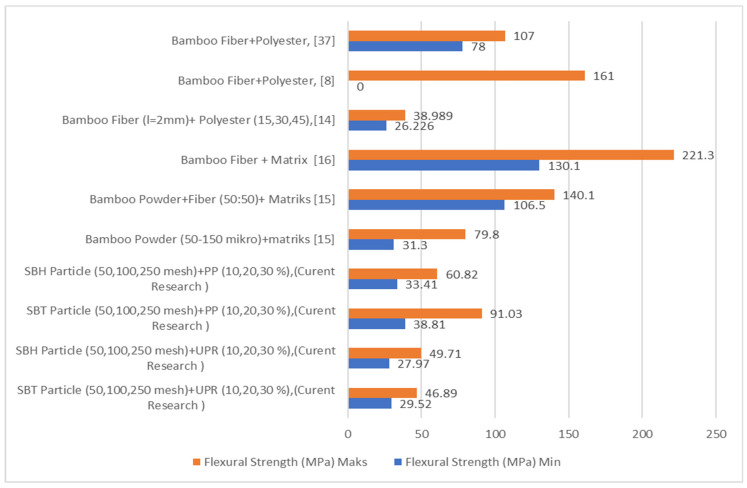
Comparison of the flexural strength of bamboo–polyester composites from previous and current research.

**Figure 9 polymers-13-04377-f009:**
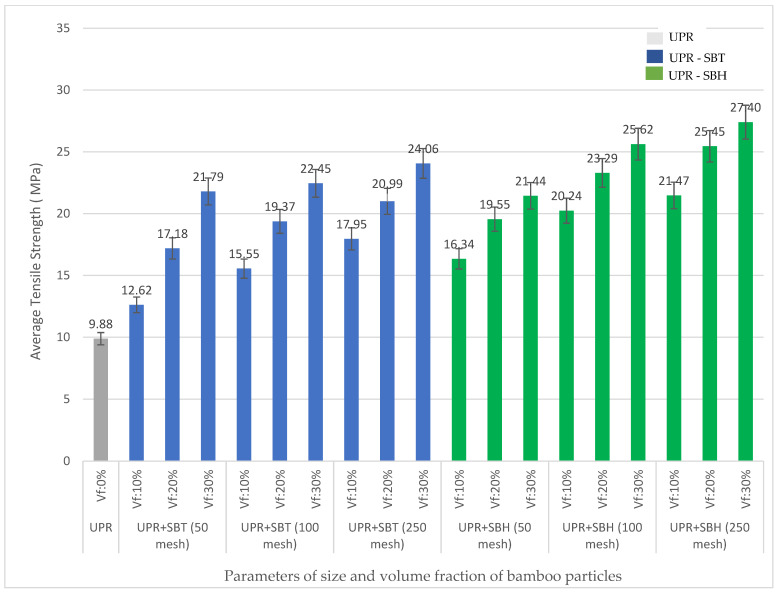
Relationship between volume fraction on the particle size of SBT and SBH polyester (UPR) matrix on tensile strength.

**Figure 10 polymers-13-04377-f010:**
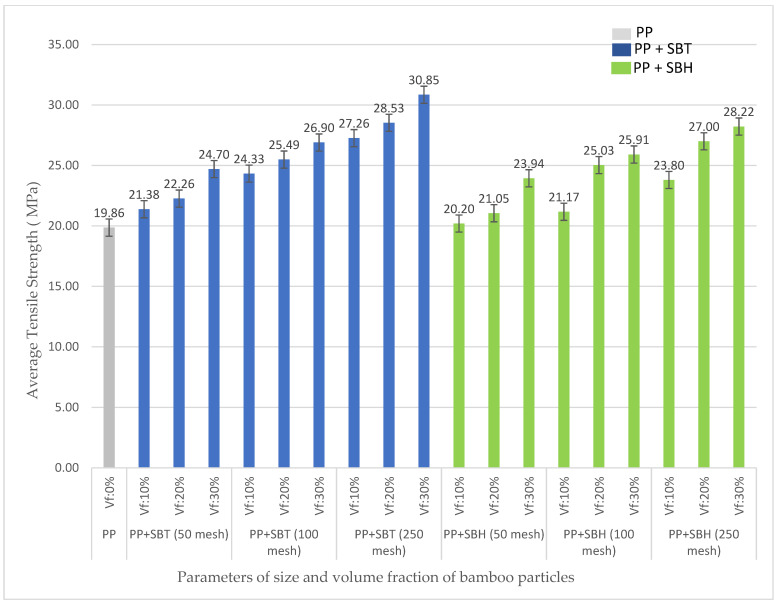
Relationship between volume fraction on the particle size of SBT and SBH polyester (PP) matrix on tensile strength.

**Figure 11 polymers-13-04377-f011:**
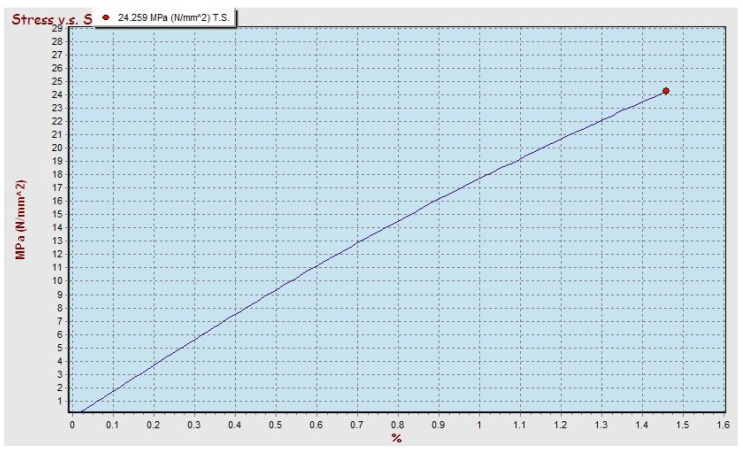
Stress–strain relationship in the tensile test of UPR–SBT composite with 30 vf% and 250 mesh particle size.

**Figure 12 polymers-13-04377-f012:**
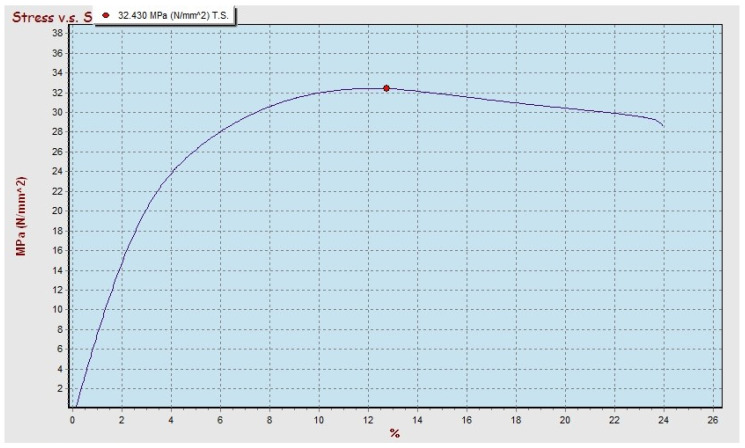
Stress–strain relationship in the tensile test of PP–SBT composite with 30 vf% and 250 mesh particle size.

**Figure 13 polymers-13-04377-f013:**
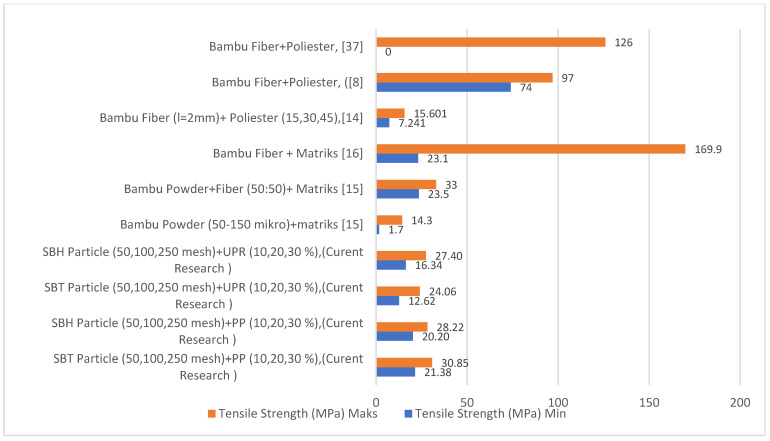
Comparison of the tensile strength of bamboo–polyester composites from previous and current research.

**Figure 14 polymers-13-04377-f014:**
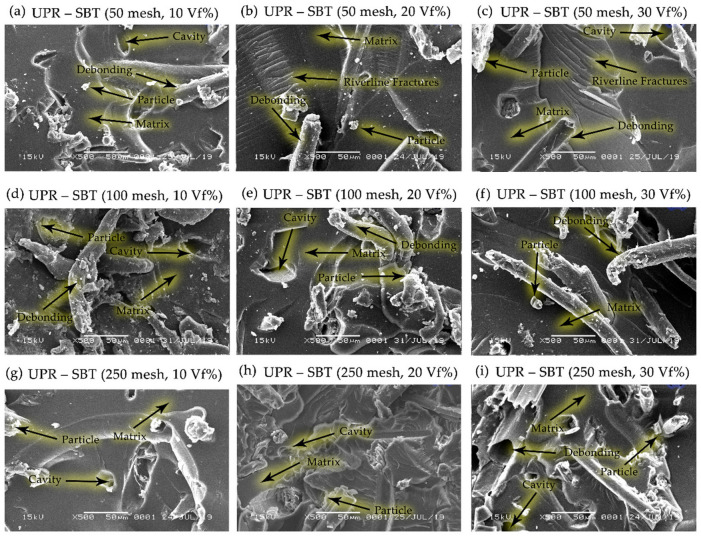
SEM micrographs on the morphology of the UPR–SBT composites.

**Figure 15 polymers-13-04377-f015:**
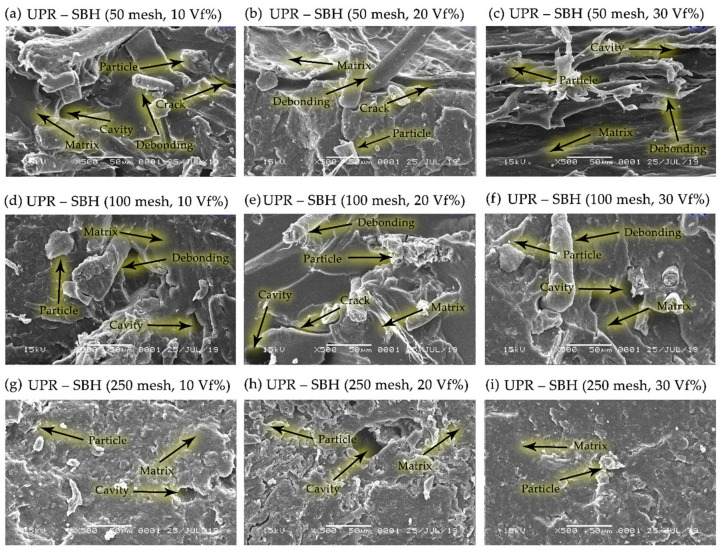
SEM micrographs on the morphology of the UPR–SBH composites.

**Figure 16 polymers-13-04377-f016:**
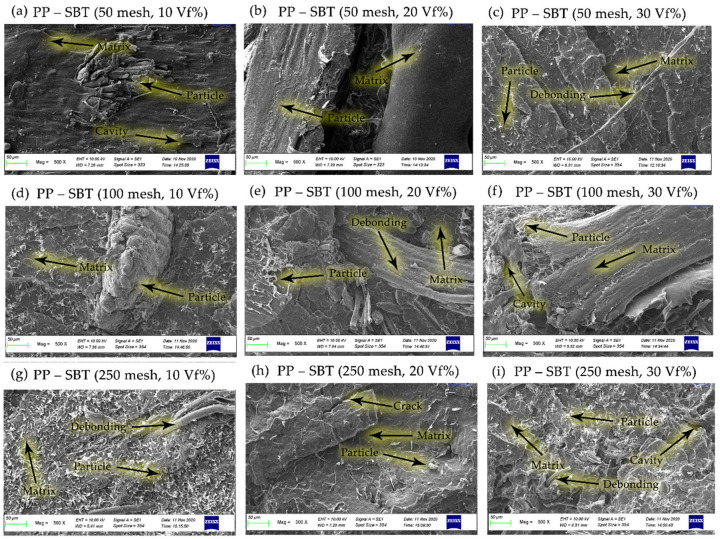
SEM micrographs on the morphology of the PP–SBT composites.

**Figure 17 polymers-13-04377-f017:**
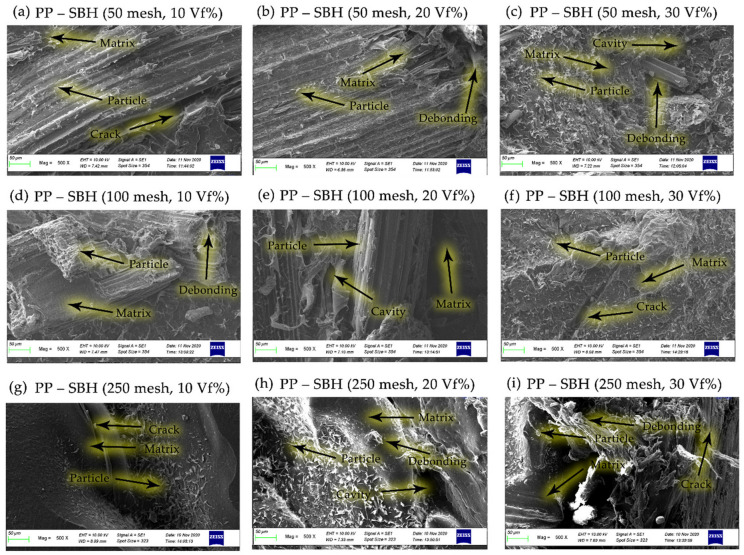
SEM micrographs on the morphology of the PP–SBH composites.

**Table 1 polymers-13-04377-t001:** Composite fabrication and characterization process parameters.

No	Process Parameters	Units	Types of Bamboo Fiber	
			SBT	SBH
1	Fiber Fraction	% volume (Vf%)	10, 20, 30	10, 20, 30
2	Fiber Size	Mesh	50, 100, 250	50, 100, 250
3	Matrix Type	Thermoset	1 polyester (UPR)	1 polyester (UPR)
		Thermoplastic	2 polypropylene (PP)	2 polypropylene (PP)
4	Composite fabrication method	Manual Machine	Hand layup Single screw extrusion-rolling	
5	Composite characterization	MPa	1 Tensile strength	
		MPa	2 Flexural strength	
		Fiber and matrix interface	3 Morphology	
6	Particleboard characterization	g/cm^3^	1 Density	
		%	2 Testing moisture content	
		%	3 Thickness expansion after immersion in water	
		MPa	4 Flexural strength in dry conditions	
		MPa	5 Flexural strength in wet conditions	
		MPa	6 Tensile strength	
		N	7 Wood screw holding power	

**Table 2 polymers-13-04377-t002:** Testing and characteristic results for particleboard.

No	Material	Particle Size (Mesh)						
		ρ (g/cm^3^)	Wab (%)	Sw (%)	σ_l-k (MPa)_	E (GPa)	σ_l-b (MPa)_	F_t-s (N)_
1	PP–SBT Composite	0.879	2.03	0.87	91.09	4.508	46.82	826
2	Polyester–SBT Composite	0.949	1.21	0.71	46.89	3.396	24.27	740
3	JIS A 5908-2003	0.4–0.9	Max 13%	Max 12%	8–30	1.3–3.0	6.5–15	500

## Data Availability

The data presented in this study are available on request from the corresponding author.

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
