# Peer review of "Polymer Composite Fabrication Reinforced with Bamboo Fiber for Particle Board Product Raw Material Application"

_polymers, 2021, doi:10.3390/polym13244377_

Round 1
Reviewer 1 Report
The paper is focused on the preparation and morphological/mechanical characterization of composite materials composed by polymers and bamboo fibers. The topic falls within the scope of the journal. The presentation and discussion of the results should be improved. On this basis, I recommend the publication after the following revisions:
- Experimental details for SEM analyses (such as energy of beam and working distance) should be reported.
- It would be interesting to present some examples of stress vs stain curves for both tensile and flexural experiments. Other mechanical properties (elastic modulus, yield point, ultimate elongation, stored energy up to breaking) could be calculated from the analysis of stress vs strain curves. The determination of additional mechanical characteristics would increase the interest of the readers to this paper.
- The scale length within SEM images (Fig. 10-13) is not clear. Please check and revise.
- In my opinion, the correlation between morphological studies and the mechanical properties of the nanocomposites should be better presented and discussed.
Author Response
Response to Reviewer 1 Comments
Point 1 :Experimental details for SEM analyses (such as energy of beam and working distance) should be reported.
Response 1:
Morphological examination using SEM analysis is equipped with data (such as energy of beam and working distance). The data is written in Figures 14, 15, 16, 17 (see the article). We provide a sign with a track change for revision from reviewer input.
Point 2 : It would be interesting to present some examples of stress vs strain curves for both tensile and flexural experiments. Other mechanical properties (elastic modulus, yield point, ultimate elongation, stored energy up to breaking) could be calculated from the analysis of stress vs strain curves. The determination of additional mechanical characteristics would increase the interest of the readers to this paper.
Response 2:
We have presented and discussed an example of a stress-strain graph on SBT-Polyester and SBT-Polypropylene composites with a volume fraction of 30% and a bamboo particle size of 30 mesh, which is a the sample that has the highest tensile strength and bending strength. Stress-strain graphs are shown in Figures 6, 7, 11 and 12 (see the article). We provide a sign with a track change for revision from reviewer input.
Point 3 :The scale length within SEM images (Fig. 10-13) is not clear. Please check and revise.
Response 3:
The scale length on SEM images were added, it can be seen in Figures 14, 15, 16 and 17 (see the article). We provide a sign with a track change for revision from reviewer input.
Point 4 : In my opinion, the correlation between morphological studies and the mechanical properties of the nanocomposites should be better presented and discussed.
Response 4:
We added the correlation between morphological studies and the mechanical properties of composites in the discussion section. Articles are marked with blue text (see the article). In this study, the reinforcement was in micron size (mesh) not in nano size. We provide a sign with a track change for revision from reviewer input.

Reviewer 2 Report
- 2.1 Materials, include information about purification materials or used as received
- Line 91, change acronym for “Polyester (PE)”, because PE is international abbreviation for polyethylene
- Why is important to include Figures 1 and 2, manuscript needs to have scientific Figures
- Manuscript has many typos, for example all manuscript has “Polyester” and correct is “polyester”. Maleic Anhydride and Polypropylene, correct are maleic anhydride and polypropylene; change “Homopolymer” for “homopolymer”, etc.
- Line 134 “Sampling for tensile and flexural tests as well as SEM”, define SEM (Scanning electron microscopy?)
- Polypropylene (PP) defined some time (lines 88 and 155), remove (PP) from line 155
- Line 197, change “10%, 20%, and 30%” for “10, 20, and 30%”
- Were difficult to understand Figures 4-9, try to prepare these Figures (scientific) for easy understand
- Figures 10-13 were no prepare according to scientific publications
- Manuscript has some interesting results but doesn’t have discussion, include discussion for all Figures
- Figure 6 has this information “SBT Powder +Polyester (10,20, 30)%,(Research Martijanti,
et.al,2020)”, include reference about this citation. Remove “Chart Title” from Figure 6.
- Manuscript has only one reference from 2020, include more recently references
- All Figures need to improve quality an also resolution at minimal 300 dpi
- The manuscript has some pictures, SEM and mechanical properties, manuscript needs to include more characterization methods
- In conclusion, manuscript has no level for publications in Polymer Journal
Author Response
Response to Reviewer 2 Comments
Point 1 : 2.1 Materials, include information about purification materials or used as received
Response 1:
2.1. Material, we explain information about purification materials or used as received. We provide a sign with a track change for revision from reviewer input. Articles are marked with blue text (see the article).
Point 2 : Line 91, change acronym for “Polyester (PE)”, because PE is international abbreviation for polyethylene
Response 2:
We wrote in the article that the abbreviation for polyester is UPR (see the article). We provide a sign with a track change for revision from reviewer input
Point 3 :Why is important to include Figures 1 and 2, manuscript needs to have scientific Figures
Response 3: :
we have redrawn the figures of the composite manufacturing process using the hand lay up and extrusion methods (see the article).
Point 4 :Manuscript has many typos, for example all manuscript has “Polyester” and correct is “polyester”. Maleic Anhydride and Polypropylene, correct are maleic anhydride and polypropylene; change “Homopolymer” for “homopolymer”, etc.
Response 4 :
The writing for polyester, polypropylene, maleic anhydride, etc, has been revised (see the article).
Point 5: Line 134 “Sampling for tensile and flexural tests as well as SEM”, define SEM (Scanning electron microscopy?)
Response 5 :
line 131, the sentence has been deleted, has been corrected by providing an explanation of the stages of the composite manufacturing process from the preparation of raw materials to producing the product.
Point 6 : Polypropylene (PP) defined some time (lines 88 and 155), remove (PP) from line 155
Response 6 :
Line 155, the sentence has been deleted. We provide a sign with a track change for revision from reviewer input. Articles are marked with blue text (see the article).
Point 7 :Line 197, change “10%, 20%, and 30%” for “10, 20, and 30%”
Response 7:
Line 197, has been revised, Articles are marked with blue text (see the article).
Point 8 :Were difficult to understand Figures 4-9, try to prepare these Figures (scientific) for easy understand
Response 8:
Figures 4 - 9 have been revised, by providing additional information on the image (see the article)
Point 9: Figures 10-13 were no prepare according to scientific publications
Response 9:
Figures 10 – 13, have been revised to improve their quality and also resolution. Morphological examination using SEM analysis is equipped with data (such as energy of beam and working distance) and The scale length within SEM images (see the article).
Point 10:Manuscript has some interesting results but doesn’t have discussion, include discussion for all Figures
Response 10:
Discussion has been revised and the discussion of the correlation between morphological studies and the mechanical properties of composites has been added . Articles are marked with blue text (see the article). We provide a sign with a track change for revision from reviewer input.
Point 11: Figure 6 has this information “SBT Powder +Polyester (10,20, 30)%,(Research Martijanti,et.al,2020)”, include reference about this citation. Remove “Chart Title” from Figure 6.
Response 11 : The current result data have not been published. We provide a sign with a track change for revision from reviewer input (see the article).
Point 12 :Manuscript has only one reference from 2020, include more recently references
Response 12: The reference has been added to the manuscript, especially reference from 2020 (see the article).
Point 13: All Figures need to improve quality and also resolution at minimal 300 dpi
Response 13:
Figure has been revised to improve quality and also resolution (see the article).
Point 14 :The manuscript has some pictures, SEM and mechanical properties, manuscript needs to include more characterization methods
Response 14:
It would be interesting to present some examples of a stress-strain graph on SBT-Polyester and SBT-Polypropylene composites with a volume fraction of 30% and a bamboo particle size of 30 mesh, in which the samples had the highest tensile strength and bending strength. Stress-strain graphs are shown in Figures 6, 7, 11 and 12 (see the article). We provide a sign with a track change for revision from reviewer input. We have added to the discussion of the correlation between morphological studies and the mechanical properties of composites.
In this study to find out the condition of bamboo species, by varying the volume fraction and bamboo particle size to obtain materials that can be applied to particle board according to JIS standards, with mechanical and morphological characteristics as well as testing on composite particle board products reinforced with bamboo particles. This study is expected to answer the problems in this research.
The words are marked with blue text (see the article). We provide a sign with a track change for revision from reviewer input (see the ).
Point 15: In conclusion, manuscript has no level for publications in Polymer Journal
Response 15:
Thank you for your assessment of this manuscript. We will try to improve this manuscript, so that it can be published in the polymer journal

Round 2
Reviewer 1 Report
The paper can be published in the present form.
Reviewer 2 Report
Manuscript is accepting in present form